# Impact of Urban Expansion on Carbon Emissions in the Urban Agglomerations of Yellow River Basin, China

Zhenwei Wang [1], Yi Zeng [2], Xiaochun Wang [1,*], Tianci Gu [2] and Wanxu Chen [2]

1   School of Public Administration, Hubei University, Wuhan 430062, China
2   Hubei Key Laboratory of Regional Ecology and Environmental Change, School of Geography and Information Engineering, China University of Geosciences, Wuhan 430074, China
*   Correspondence: wangxc@hubu.edu.cn

**Abstract:** Continued urban expansion (UE) has long been regarded as a huge challenge for climate change mitigation. However, much less is known about how UE affects carbon emissions (CEs), especially in the urban agglomerations of the Yellow River Basin (UAYRB), China. In this regard, this study introduced kernel density analysis, the Gini coefficient, and Markov chains to reveal the UE patterns and carbon emissions intensity (CEI) in the UAYRB at the county level, and explored the spatial heterogeneity of the impact of UE on CEI with the geographically and temporally weighted regression model. The results show that both CEI and UE in the UAYRB showed a steady growing trend during the study period. The kernel density of CEI and UE revealed that CEI in the UAYRB was weakening, while the UE rate continuously slowed down. The Gini coefficients of both CEI and UE in the UAYRB region were at high levels, indicating obvious spatial imbalance. The Markov transfer probability matrix for CEI with a time span of five years showed that CEI growth will still occur over the next five years, while that of UE was more obvious. Meanwhile, counties with a regression coefficient of UE on CEI higher than 0 covered the majority, and the distribution pattern remained quite stable. The regression coefficients of different urban landscape metrics on CEI in the UAYRB varied greatly; except for the landscape shape index, the regression coefficients of the aggregation index, interspersion and juxtaposition index, and patch density overall remained positive. These findings can advance the policy enlightenment of the high-quality development of the Yellow River Basin.

**Keywords:** urban expansion; carbon emissions; landscape pattern index; geographically and temporally weighted regression; urban agglomerations; Yellow River Basin; China

## 1. Introduction

Global warming is a huge challenge facing humankind [1]. Greenhouse gases (GHGs) generated by human activities are the culprit [2]. It is reported that 2% of global urban areas generate approximately 75% of the global carbon emissions (CEs). The past century has witnessed a great change in urban population growth and urban expansion worldwide [3]. From 2000 to 2010, global urban areas grew by an average of 5694 km$^2$ per year, resulting in a net loss of 22.4 Tg carbon per year [4]. Urban expansion (UE) can also indirectly affect carbon stocks, which is difficult to fully quantify [5]. Research found that, from 1985 to 2015, approximately 12% and 9% of UE came at the expense of grassland and forest, respectively, while UE in China is expected to create a 1800 km corridor of coastal cities from Hangzhou to Shenyang [3]. China has been experiencing unprecedented urbanization since economic liberalization began in 1978 [6]. According to the National Bureau of Statistics, in 2015, China's urban built-up area was 1.6 times that in 2005, reaching 52,102 km$^2$. The nonnegligible eco-environmental issues caused by continuing UE has concerned the global academics [7–9]. Urban agglomerations will be the main front of UE since it remains the primary form of urbanization until at least 2035 [10]. As the strategic core area of national

economic development, urban agglomerations shoulder the historical responsibility of carrying the shift in the world's economic center of gravity [11,12]. Although agglomeration has promoted economies of scale and facilitated better infrastructure and services, the development of urban agglomerations has also been accompanied by unprecedented energy consumption, leading to growing challenges in eco-environmental issues related to climate change [13–16]. Owing to intensive human socioeconomic activities, urban agglomerations are inevitably becoming the major generators of CEs [10]. However, the spatial relationship between UE and CEs has not been comprehensively and thoroughly examined. Existing research has mostly focused on the impact of UE on CEs of city individuals, while studies are rare at the scale of urban agglomerations from the perspective of urban landscape metrics [17].

Currently, research is substantial concerning UE, and has mainly focused on the following aspects. (1) Characterization of UE. Quantitative indexes, such as expansion intensity and expansion direction, have been widely used to characterize the UE pattern [18,19]. Meanwhile, the extensive use of 3S has facilitated better understanding the spatiotemporal evolution of UE [20]. Jiao et al. (2018) proposed a new landscape metric to characterize the evolution process of UE and observed an increasingly decentralized spatial pattern [21]. (2) UE simulation. Since UE is a prevailing phenomenon worldwide, whether it exhibits some particular features arouses the attention of scholars. Seto et al. (2012) found that urban areas will increase by 1.2 million km$^2$ by 2030 under the current population density trend [3]. Guo et al. (2022) introduced the patch-based land use simulation (PLUS) model to simulated UE of the Harbin–Changchun urban agglomerations under ecological constraints and found that the PLUS model can better simulate UE at the scale of urban agglomerations [22]. Nevertheless, although exhibiting similar population or economic growth, countries will vary in the probabilities of UE. Seto et al. (2012) found that UE likelihood in individual countries tends to exhibit both high spatial variability and high spatial concentration, while some countries with low probabilities of UE show a high uncertainty of expansion pattern, such as Turkey [3]. (3) Drivers of UE. Natural, socioeconomic, and political factors are widely believed to be the traditional driving factors of land use change [23]. However, the unified analysis framework of drivers of urban land expansion has not reached a consensus [22]. Meanwhile, factors affecting UE vary greatly among regions. For example, the leading contributor to UE in China is GDP, while that in India and Africa is population [18]. (4) Impacts of UE on the eco-environment. The impacts of UE on the eco-environment can be divided into two perspectives of direct and indirect impacts. UE can directly affect natural habitats by converting them to urban use [24]. Mao et al. (2018) revealed that urbanization-induced wetland loss reached 2883 km$^2$ from 1990 to 2010 [25]. Among them, economically developed urban agglomerations are the hotspots of urbanization-induced wetland loss in China. Furthermore, UE from 1992 to 2016 has resulted in an average 0.8% loss of dryland habitat quality [26]. Liu et al. (2019) conducted research at the global scale to reveal the impacts of UE on terrestrial net primary productivity (NPP) and found that global terrestrial NPP loss equaled ~9% of the CEs from fossil fuel and cement emissions worldwide [4]. Although research is substantial on the direct impact of UE on the eco-environment, the indirect impact of UE on natural habitats is more severe than the direct one [9]. It is estimated that cropland expansion contributes the greatest to natural area losses globally, while the indirect impact of UE on natural area losses is significantly underestimated [9]. Ren et al. (2022) found that dryland UE has indirectly affected nearly 60% of threatened species [26]. We can conclude that the environmental impacts of UE have been extensively studied, while studies on the impacts of UE on CEs still leave much to be carried out, especially at the urban agglomerations level as they have become the major driver of CEs.

With the rapid development of urbanization in China, the impact of UE on CEs is one of the current research hotspots in the field of environment and economy [27–30], and understanding the impact of UE on CEs is crucial to the formulation of effective low-carbon development policies. However, little is known about how UE affects CEs, especially in

the UAYRB. To this end, this study introduced kernel density, the Gini coefficient, and Markov chains to reveal the UE and CEI patterns in the UAYRB at the county level, and explored the spatial heterogeneity of the impact of UE on CEI with the geographically and temporally weighted regression (GTWR) model, and this provides a new perspective and methodological framework for urban CEs research. The impact of UE on CEs in the UAYRB in China has profound economic implications and practical significance. First, understanding the correlation between UE and CEs allows for more efficient resource allocation. Secondly, encouraging low-carbon sustainable urban development can promote the development of green technologies and industries, which, in turn, can create new jobs and promote economic growth. In terms of policymaking, identifying the impact of UE on CEs can provide decision makers with information that can help them to formulate effective urban planning and environmental policies to curb CEs while managing UE. This study theoretically explores and analyzes the impact of UE on CEs to provide new ideas for the theory of urban development and low-carbon transition. In the past, related studies lacked an exploration of the UAYRB as a specific region, and, by revealing the dynamic impact of UE on CEs, we provide experience and an important reference for CEs reduction studies in similar regions. And, methodologically, the kernel density analysis and Gini coefficient proposed for use in this study provide theoretical support and methodological exploration for further research in this area.

Studies on the impacts of UE on CEs are not scarce. Krayenhoff et al. (2018) found a nonlinear interaction between GHG-induced warming and corresponding UE in American cities [8]. Liu and Zhang (2022) found that the positive trade-offs between UE and ecological construction could mitigate CEs growth in China's urban agglomerations [10]. Cheng et al. (2022) revealed that cities with a larger population tend to have lower per capita CEs [31]. Actually, compared with low-density communities, high-density communities tend to have lower per capita energy use [32,33]. Urban population expansion is usually accompanied by urban area expansion, but an easily overlooked fact is that urban areas around the world are expanding twice as fast as their populations on average [18,34]. Much has been carried out to curb such expansion. However, contrary to what we believe—that land use planning is an effective way to curb UE [35]—it actually stimulates fragmented UE [36], though it is regarded as uneconomic, inefficient, and environmentally unfriendly [7,37]. The landscape pattern index can well characterize the UE pattern, which can facilitate better understanding the impacts of UE on CEs.

How urban area expansion affects CEs in the urban agglomerations area with the rapid development of urbanization remains an urgent issue that requires a prompt solution, most notably regarding urban agglomerations of the Yellow River Basin (UAYRB). As an important ecological barrier, food base, and economic zone in China, the Yellow River Basin concentrates a large amount of chemical, energy, and production industries, making it an ecologically fragile region with a high concentration of CEs and pollution [38]. The inherent problems of unbalanced development and unfriendly ecological environment in the Yellow River Basin require being solved by the development of urban agglomerations from point to area [39]. Within this context, this research intends to address the gaps mentioned above by proposing the following research objectives. (1) What are the UE patterns and CEs intensity (CEI) in the UAYRB? (2) What is the spatial heterogeneity of the impact of UE on CEI in the UAYRB?

## 2. Materials and Methods

### 2.1. Study Area

From west to east, the Yellow River flows through 9 provinces and regions, with a total length of 5464 km and an area of 795,000 km$^2$, accounting for 8.28% of the country's total. It is the second longest river in China and the fifth longest in the world. The Yellow River and its coastal basin are among the most important birthplaces of the Chinese nation, and they are also the main battlefield of national environmental civilization construction. In 2021, the State Council of the Central Committee of the Communist Party of China

upgraded the ecological protection and high-quality development of the Yellow River Basin to a major national strategy, highlighting the strategic position of the Yellow River Basin in the overall situation of national development and socialist modernization [39]. The UAYRB consists of 7 urban agglomerations, Shandong Peninsula Urban Agglomerations (SPUA), Central Plains Urban Agglomerations (CPUA), Guanzhong Plain Urban Agglomerations (GPUA), Lanxi Urban Agglomerations (LXUA), Jinzhong Urban Agglomerations (JZUA), Hohhot–Baotou–Ordos–Yulin Urban Agglomerations (HBOYUA), and Ningxia Urban Agglomerations (NXUA) along the Yellow River (Figure 1). The seven urban agglomerations account for about 33.6% of the area of the Yellow River Basin, and the proportion of the population and main economic indicators in China's urban agglomerations is roughly between 20% and 25%. At the same time, that of the nine provinces and regions in the Yellow River Basin is as high as 60~70% [40]. It is a high-density population-gathering area in the Yellow River Basin, an important area for high-quality economic development, a heavy-loaded area for inheriting the Yellow River culture and promoting Chinese civilization, and a key area for comprehensive environmental pollution control and ecological protection. Therefore, it has a very important strategic position in the high-quality development of the Yellow River Basin.

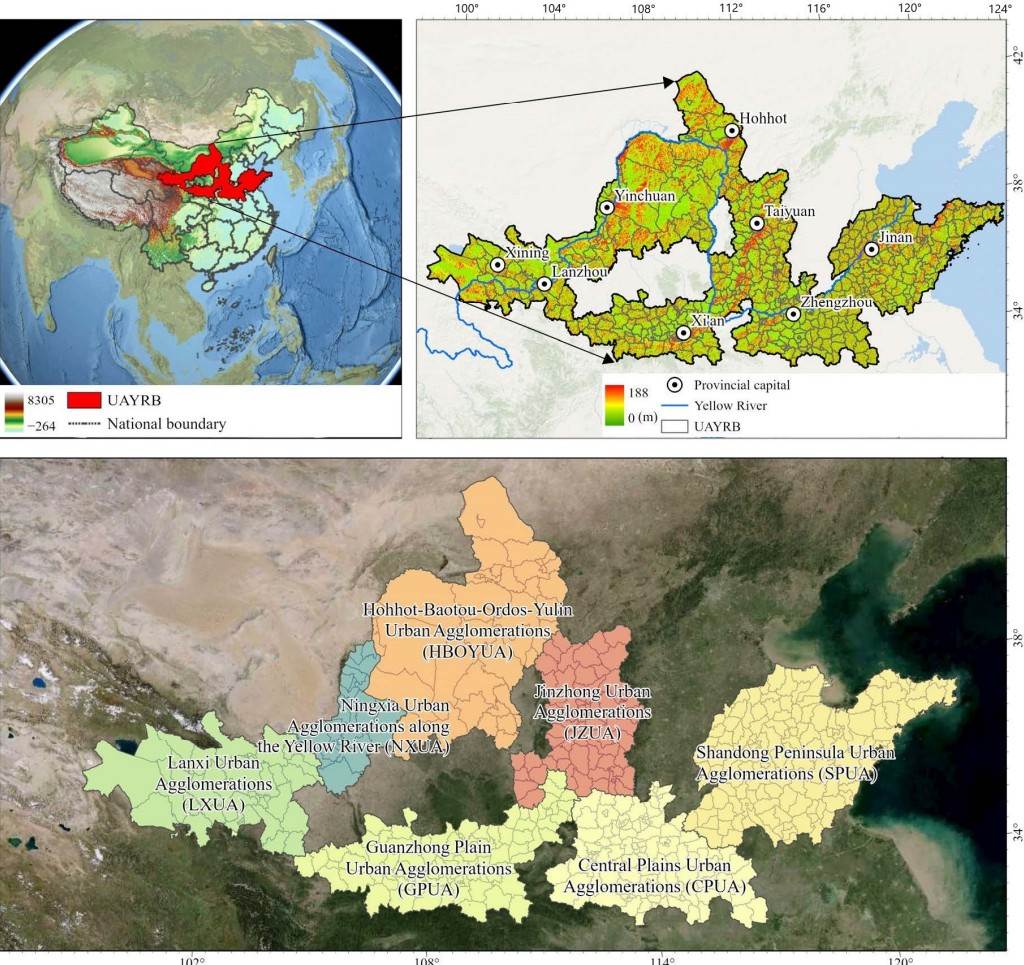

**Figure 1.** Study area.

## 2.2. Data Sources

The data in this study involved CEs and land use data. CEs data were sourced from the open-source data inventory (https://db.cger.nies.go.jp/dataset/ODIAC/, accessed on 7 May 2024) [41]; these CEs data successfully estimate the spatial distribution of fossil fuel CEs on a global scale by combining night-time lighting data and emission location

profiles of individual power plants using an innovative emission modeling approach. The spatial resolution is 1000 m. The land use change data were downloaded from the Data Center for Resources and Environmental Sciences and the Chinese Academy of Sciences (https://www.resdc.cn/, accessed on 7 May 2024). The spatial resolution is 30 m. This study used the proportion of urban land and the landscape pattern index to characterize UE [42]. The proportion of urban land is the ratio of urban land area to the area of the study unit, and the landscape pattern indexes were calculated in Fragstats v4.2.1 (Oregon State University, Corvallis, OR, USA).

*2.3. Methods*

This study aims to analyze the impact of UE on CEs. We adopt a series of models to assess the association between them to achieve this goal. Through these models (Figure 2), we expect to reveal the impact of UE on CEs and provide a scientific basis for formulating low-carbon development policies. To demonstrate the impact of UE on CEs in the UAYRB, this study uses the urban land area share and the landscape pattern index to characterize the spatiotemporal changes of UE, and, at the same time, we use the CEI as an indicator to analyze the changes in CEs. By calculating these indicators, we use kernel density analysis to reveal the dynamic change patterns of UE and CEI and analyze the imbalance of UE and CEI based on the Gini coefficient. In addition, we apply Markov chains to predict future changes in the trends of UE and CEI and reveal the impact of UE on CEI through the GTWR model. The core analytical approach of this study focuses on showing the changing patterns of UE and CEs in the UAYRB and further exploring the mechanism of UE's impact on CEI.

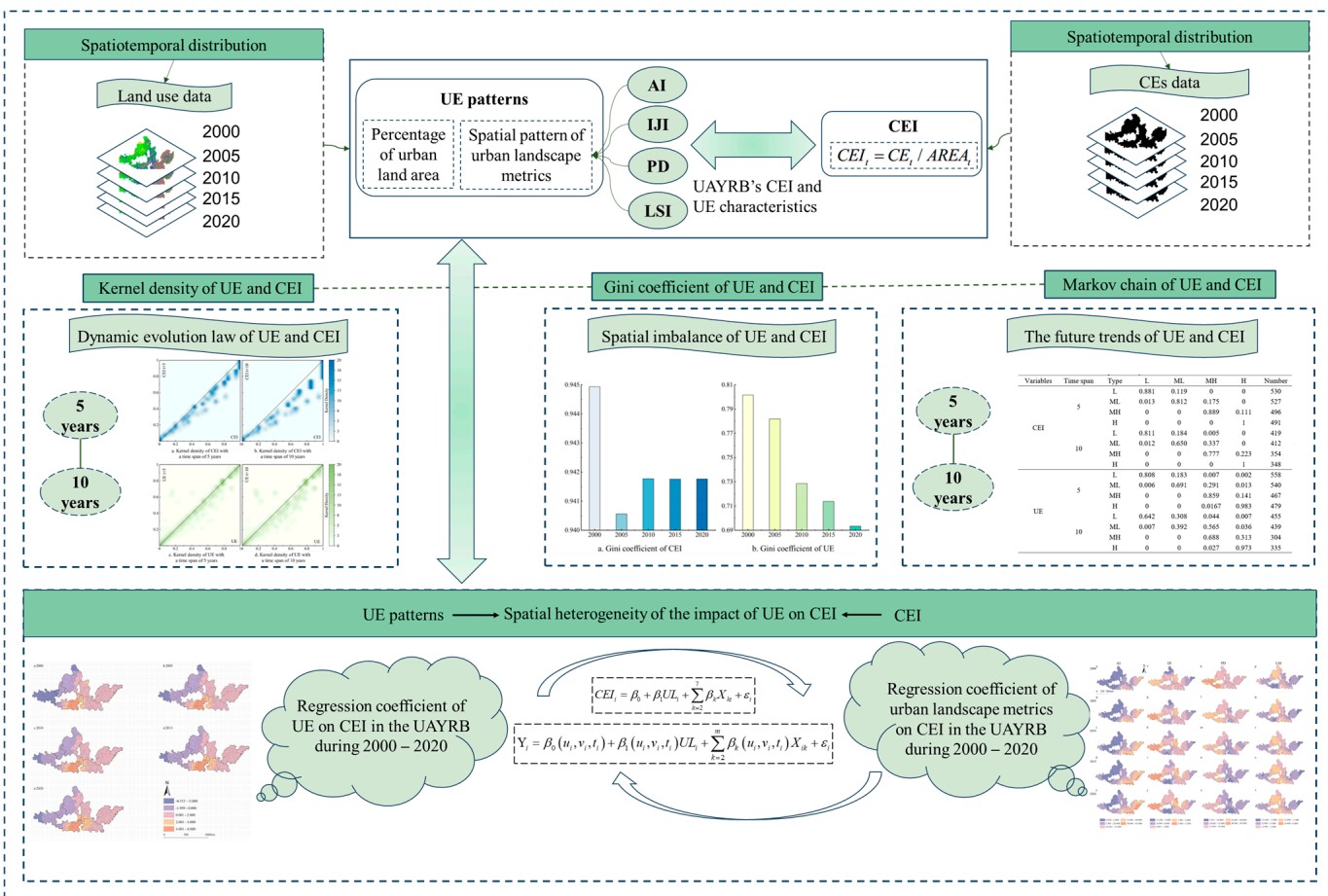

**Figure 2.** Basic flowchart of the method.

### 2.3.1. UE Measurement

This study uses the proportion of urban land to visualize the spatial pattern and changes in UE. We also introduced four landscape metrics—aggregation index (AI), interspersion and juxtaposition index (IJI), patch density (PD), and landscape shape index (LSI)—to describe the urban landscape in the UAYRB [43–45]. The AI examines the connectivity between patches of each landscape type and is used to measure the degree of aggregation of urban land. The IJI is used to assess the degree of interlocking and juxtaposition between different land types. PD expresses the density of urban land in the landscape, which can reflect the heterogeneity and fragmentation of urban land. The LSI, as a landscape shape index, reflects the changes in the shape of land use patches in the process of urbanization, and a high LSI value may indicate that UE has led to a more complex shape of land use patches. These indicators can visualize the changing characteristics of UE in the UAYRB.

### 2.3.2. CEI Assessment

The measurement of CEI in this study is expressed as the ratio of total CEs to area for each unit [46], and the unit is t $CO_2/km^2$. The calculation equation is as follows:

$$CEI_t = CE_t / AREA_t \tag{1}$$

where $CEI_t$ is the CEs intensity in year $t$; $CE_t$ is CEs in year $t$; and $AREA_t$ is the area of administrative units.

### 2.3.3. Kernel Density

Kernel density estimation is a nonparametric estimation, which has the advantage of not relying on sample data and has been widely used by scholars [47,48]. The dynamic evolutionary trend of the research elements is estimated by the density function to reveal its changing rules. Here, we introduced the method of kernel density estimation to explore the dynamic evolution law of UE and CEI in the UAYRB. We used the Gaussian function as the kernel function and referred to Yang et al. (2023a) to calculate the kernel density analysis [49].

### 2.3.4. Gini Coefficient

The Gini coefficient is often used to measure the degree of imbalance and insufficiency of regional economic development [50]. This study introduced the Gini coefficient to analyze the CEI and the spatial imbalance of the UE of the UAYRB. The Gini coefficient is applied to assess the uneven spatial distribution of UE and CEI, thus revealing the uneven spatial distribution of UE on CEI. It can provide us with a new perspective to understand the relationship between UE and CEI.

### 2.3.5. Markov Chain

Markov chains are characterized by non-aftereffects, analyzing the transfer patterns based on the current state of change in the study elements and thus predicting the future trend of change [49,51]. Markov chains are introduced to analyze the trend of interconversion between different orders of magnitude by constructing Markov transfer probability matrices. Here, we introduced Markov chains to analyze the future trends of UE and CEI in the UAYRB. Markov chain analysis can help us understand the spatiotemporal evolution of UE and thus predict the future trend of CEI. By analyzing the Markov chain of UE, we can reveal the long-term impact of UE on CEI and provide a scientific basis for future urban planning and CEI control.

2.3.6. GTWR Model

This study employed the global regression (ordinary least squares method, OLS) model without considering spatial factors to investigate the impact of UE on CEI in Chinese urban agglomerations. The model was constructed as follows:

$$CEI_i = \beta_0 + \beta_1 UL_i + \sum_{k=2}^{7} \beta_k X_{ki} + \varepsilon_i \tag{2}$$

where $i$ represents a district or county, $CEI_i$ represents the CEs intensity value of a district or county $i$, $UL_i$ represents the $UL$ of district or county $i$, $X_{ki}$ represents the urban landscape pattern index affecting the CEI of district or county $i$, and $\varepsilon_i$ is the residual term.

The changes in urban land and CEI are panel data with multiple time series, while changes in urban land do not immediately cause changes in CEI, and their effects may have some lag effect [52]. The GWR only considers the spatial relationship of the cross-sectional data at a single time, which is insufficient for studying time-series data. GTWR solves this problem. The equation is as follows:

$$Y_i = \beta_0(u_i, v_i, t_i) + \beta_1(u_i, v_i, t_i)UL_i + \sum_{k=2}^{m} \beta_k(u_i, v_i, t_i)X_{ik} + \varepsilon_i \tag{3}$$

where $(u_i, v_i, t_i)$ is the sample point with spatial coordinates and timestamps, $m$ is the number of samples, $\varepsilon_i$ is the random error term, and $\beta_k$ is the estimated local regression coefficients. To make the time-series variation more apparent, in this study, the regression results of every five years are averaged to compare the changes in driving mechanisms on a ten-year scale.

**3. Results**

*3.1. CEI in the UAYRB*

The CEIs of the UAYRB in 2000, 2005, 2010, 2015, and 2020 were 10.131, 18.191, 26.816, 30.991, and 32.712 t $CO_2$/km$^2$, respectively, showing a steady growing trend during the study period. The highest CEI in 2020 reached 1732.712 t $CO_2$/km$^2$, almost three times as much as that in 2000. However, the mean CEI of the UAYRB is no more than 40 t $CO_2$/km$^2$, with the highest value of only 32.712 t $CO_2$/km$^2$ in 2020. The standard deviation of CEI in the study period exhibited a rocketing upward trend, indicating an obvious spatial difference in the CEI of the UAYRB. Spatially, the high CEI was concentrated in the SPUA and CPUA and scattered in the GZUA and JZUA during the study period (Figure 3). As time went by, the CEI in these areas showed an increasing trend, while the proportion of high CEI increased accordingly. The SPUA and CPUA showed the most obvious change characteristics. The regions with low CEI west of the HBOYUA, west and the north of the LXUA, and south of the NXUA exhibited no evident changes during the study period.

*3.2. UE Patterns in the UAYRB*

The proportions of urban land in the UAYRB in 2000, 2005, 2010, 2015, and 2020 were 6.7%, 7.8%, 10.3%, 10.8%, and 11.4%, respectively, showing an obvious increasing trend during the study period (Figure 4). Similar to the spatial pattern of CEI, the regions with a high proportion of urban land were concentrated in the SPUA and CPUA. Meanwhile, some major cities of the other urban agglomerations also have a high proportion of urban land. Over time, the spatial pattern of the regions with a high proportion of urban land showed a westward expansion trend. The SPUA and CPUA were still urban agglomerations with the most significant changes. Compared to the study period of 2000–2005, 2005–2020 witnessed the proportion of urban land increasing more evidently.

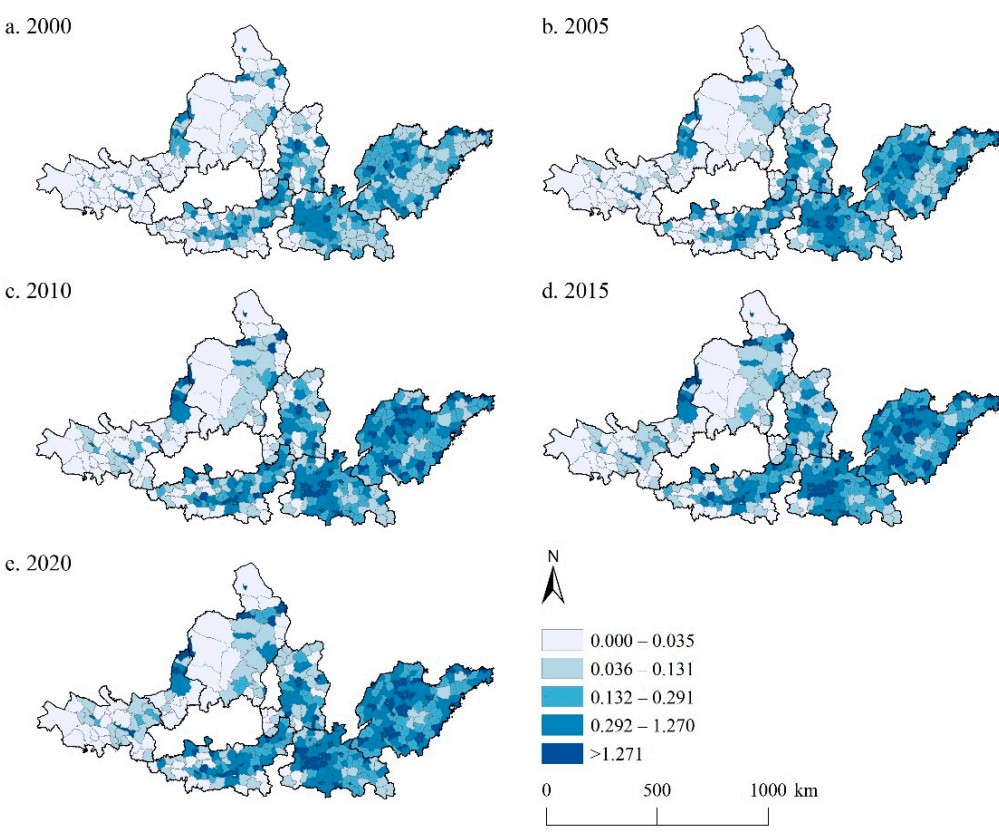

**Figure 3.** CEI in the UAYRB during 2000–2020. (**a**–**e**) are the CEI in the UAYRB in 2000, 2005, 2010, 2015, and 2020, respectively.

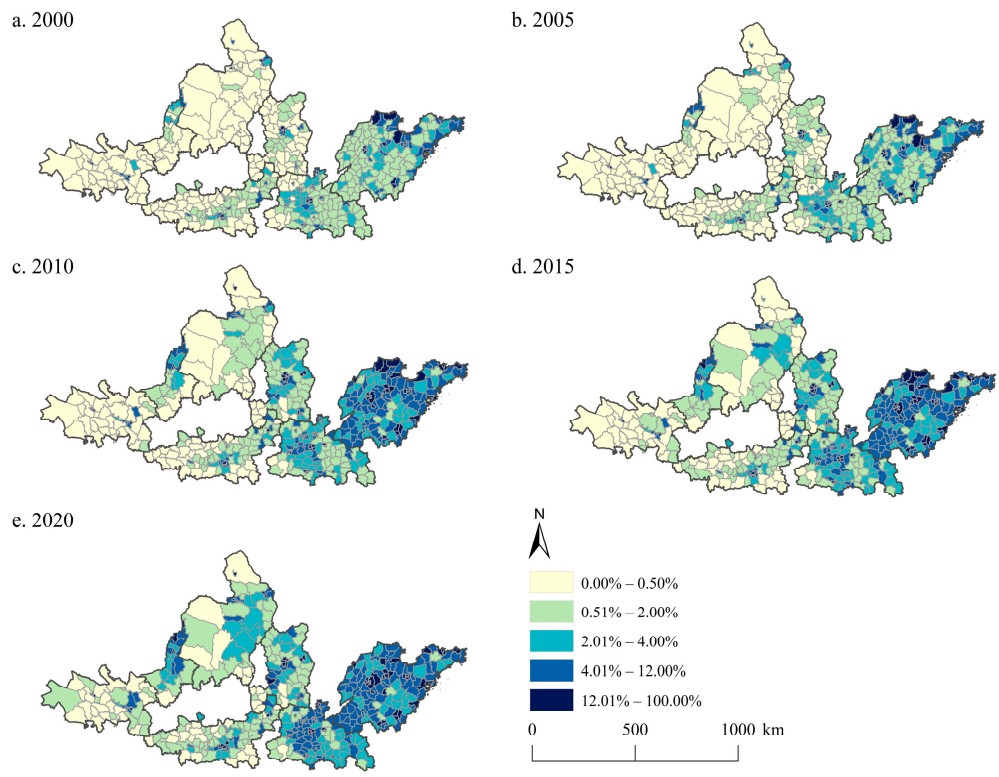

**Figure 4.** UE patterns in the UAYRB during 2000–2020. (**a**–**e**) are the UE pattern in the UAYRB in 2000, 2005, 2010, 2015, and 2020, respectively.

To better reveal the spatial pattern of UE, we introduced four landscape metrics of AI, IJI, PD, and LSI to describe the urban landscape in the UAYRB (Figure 5). Statistically, the AI in the UAYRB in 2000, 2005, 2010, 2015, and 2020 was 96.648, 96.655, 96.244, 96.284, and 95.982, respectively, exhibiting an overall declining trend but remaining relatively stable during the study period. Spatially, an AI greater than 90 covered the vast majority of regions, while an AI greater than 96 shrunk quite obviously over time, especially in the west of the UAYRB. The counties with a high IJI were most concentrated in the HBOYUA, LXUA, and NXUA, while the SPUA and CPUA clustered in low -IJI counties. Notably, the spatial pattern of a high IJI showed an eastward expansion trend over time. PD in the UAYRB in 2000, 2005, 2010, 2015, and 2020 was 0.026, 0.029, 0.052, 0.054, and 0.065, respectively, showing a steady growing trend. Spatially, SPUAs remained the regions with higher PD, while counties with high PD tended to move west over time. The LSI in the UAYRB in 2000, 2005, 2010, 2015, and 2020 was 2.112, 2.286, 3.854, 4.046, and 4.404, respectively, also showing a growing trend. However, the spatial distribution pattern of the LSI was quite different from the former three. In 2000 and 2005, an LSI lower than 15 covered almost the whole UAYRB. As time passed, counties with a high LSI mainly occurred in the HBOYUAH and the JZUA.

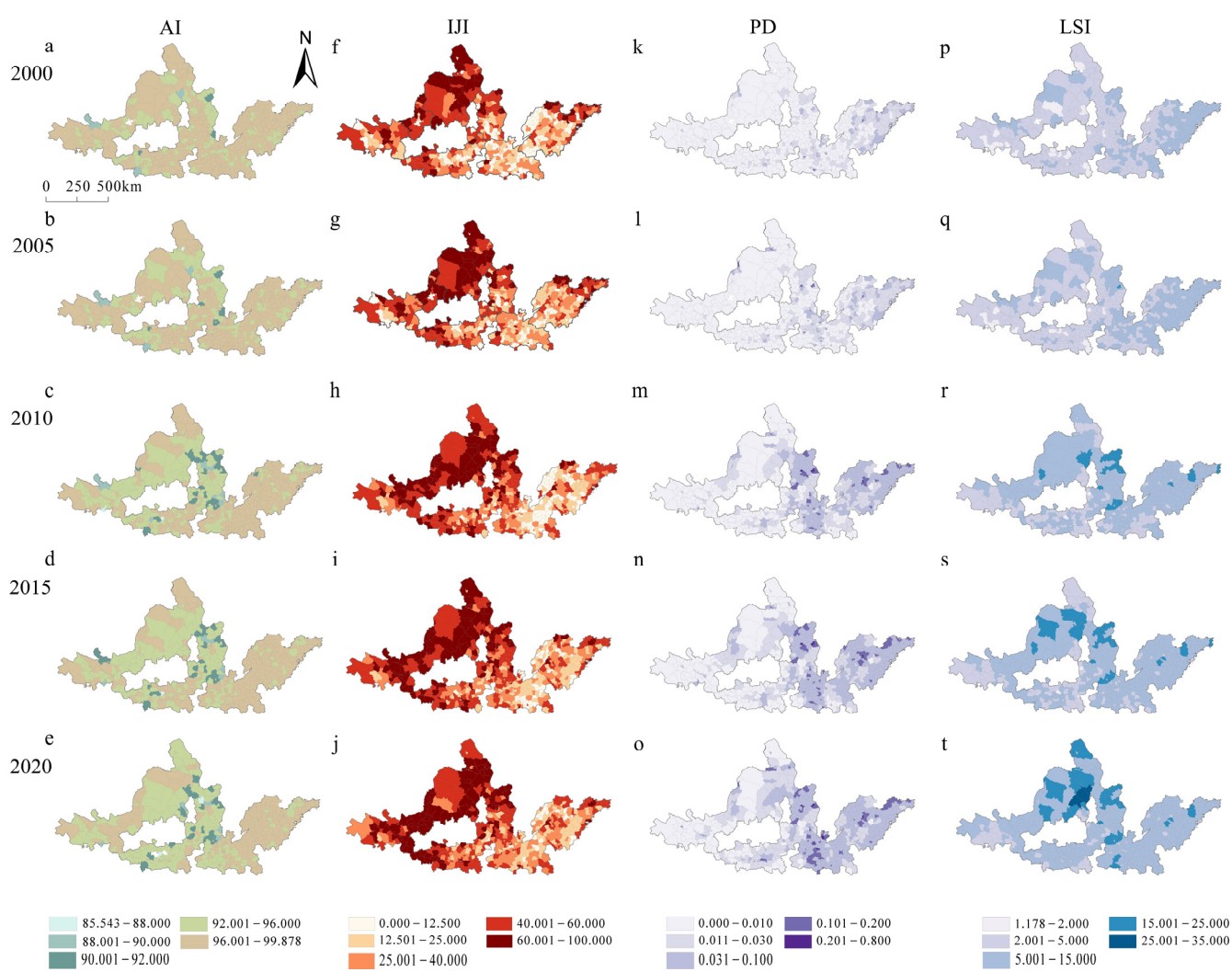

**Figure 5.** Spatial pattern of urban landscape metrics in the UAYRB during 2000–2020.

### 3.3. Kernel Density of CEI and UE

The kernel density contours of the normalized CEI and UE are shown in Figure 6. If the position of the contour lines is located near the 45° diagonal, it indicates that the

study elements did not change drastically during the study period. Figure 6a and 6b show the kernel density results for CEI over a 5-year and 10-year period, respectively. It was found that the region of high values of CEI was continuously shifting to lower values, which also indicates the weakening of CEI in the UAYRB. Furthermore, this was further corroborated by the fact that the CEI shifted sharply to lower values over time. Figure 6c and 6d show the trend of UE transfer over a 5-year and 10-year time span, respectively. The results show that the kernel density contour peak was located near 45°, indicating the basic stability of the UE and the continuous slowing down of the UE rate in the UAYRB region. However, it is worth noting that, in some of the lower-value areas, a significant shift in UE to higher values may occur, which suggests that the potential for large-scale UE still exists in some counties.

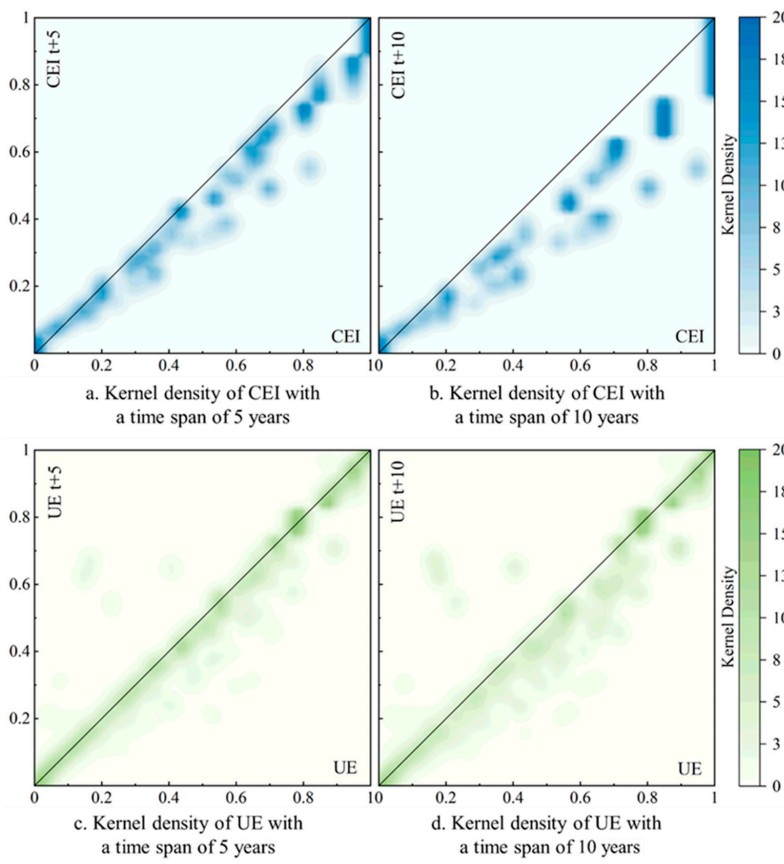

**Figure 6.** Kernel density contour map of CEI and UE. (**a**) is the kernel density of CEI with a time span of 5 years. (**b**) is the kernel density of CEI with a time span of 10 years. (**c**) is the kernel density of UE with a time span of 5 years. (**d**) is the kernel density of UE with a time span of 10 years.

### 3.4. Gini Coefficient of CEI and UE

The Gini coefficient of CEI and UE in the UAYRB during the study period is shown in Figure 7. The result of the Gini coefficient reveals the imbalance of CEI and UE. Generally speaking, the Gini coefficients of both CEI and UE were at high levels, indicating an obvious spatial imbalance of CEI and UE in the UAYRB region. As for the change in the Gini coefficient of CEI (Figure 7a), the CEI generally showed a downward trend, but the change was not dramatic. The Gini coefficient of CEI decreased by 3.357% during the study period, indicating that a slight decrease in the spatial imbalance of CEI occurred. The change in the Gini coefficient of UE showed an overall decreasing trend, and its decrease was greater than that of CEI, which is the same as in previous studies. Specifically, the decline in UE amounted to 13.506%. Urban land expansion has gradually taken on a regionally balanced pattern, partly attributed to China's strategy of balanced regional development.

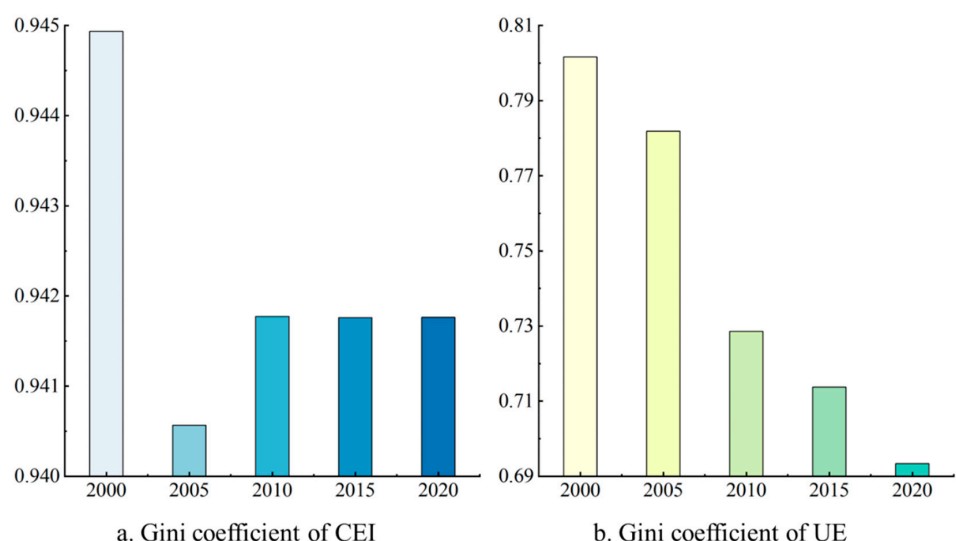

**Figure 7.** Gini coefficient of CEI and UE.

### 3.5. Markov Chain of CEI and UE

Markov chains are used to predict the future tendency of CEI and UE, and the constructed Markov transfer probability matrix is shown in Table 1. We broke CEI and UE into L (Low), ML (Middle-Low), MH (Middle-High), and H (High) using quartiles as interval points. The Markov transfer probability matrix for CEI with a period of five years showed that CEI growth will still occur over the next five years. However, when combined with the Markov transfer probability matrix for CEI with a period of 10 years, this tendency to converge to higher values kept weakening. This demonstrates the slowing growth rate of CEI as well. From the Markov transfer probability matrix of UE, the tendency to converge to higher values was more obvious than that of CEI for the five years, indicating the existence of the probability of larger-scale UE still occurring in the next five years. As the time span increases, the likelihood of UE shifting to higher values increases further, indicating that the UE remained in an increasing process.

**Table 1.** Markov transfer probability matrix of CEI and UE.

| Variables | Time Span | Type | L | ML | MH | H | Number |
|---|---|---|---|---|---|---|---|
| CEI | 5 | L | 0.881 | 0.119 | 0 | 0 | 530 |
| | | ML | 0.013 | 0.812 | 0.175 | 0 | 527 |
| | | MH | 0 | 0 | 0.889 | 0.111 | 496 |
| | | H | 0 | 0 | 0 | 1 | 491 |
| | 10 | L | 0.811 | 0.184 | 0.005 | 0 | 419 |
| | | ML | 0.012 | 0.650 | 0.337 | 0 | 412 |
| | | MH | 0 | 0 | 0.777 | 0.223 | 354 |
| | | H | 0 | 0 | 0 | 1 | 348 |
| UE | 5 | L | 0.808 | 0.183 | 0.007 | 0.002 | 558 |
| | | ML | 0.006 | 0.691 | 0.291 | 0.013 | 540 |
| | | MH | 0 | 0 | 0.859 | 0.141 | 467 |
| | | H | 0 | 0 | 0.0167 | 0.983 | 479 |
| | 10 | L | 0.642 | 0.308 | 0.044 | 0.007 | 455 |
| | | ML | 0.007 | 0.392 | 0.565 | 0.036 | 439 |
| | | MH | 0 | 0 | 0.688 | 0.313 | 304 |
| | | H | 0 | 0 | 0.027 | 0.973 | 335 |

Notes: L denotes low-level type; ML denotes medium-low-level type; MH denotes medium-high-level type; H denotes high-level type.

### 3.6. Spatial Heterogeneity of the Impact of UE on CEI

The regression coefficient of UE on CEI in the UAYRB in 2000, 2005, 2010, 2015, and 2020 was 1.111, 1.149, 1.210, 1.277, and 1.382, respectively, showing an increasing trend (Figure 8). Overall, counties with regression coefficients above 0 covered the majority, maintaining a relatively stable distribution pattern. Although areas with regression coefficients lower than 0 also exhibited similar stable distribution patterns, mainly concentrated west of the UAYRB, the proportion of the lowest regression coefficient shrunk to a few counties in the west of LXUA over time. The regression coefficient in the whole LXUA and NXUA, the majority of areas of HBOYUA, remained negative during the study period. By comparison, the regression coefficient in the JZUA and the SPUA was always positive. The regression coefficient stayed between 0.001 and 2.000. In 2020, the regression coefficient of two counties in the CPUA shifted from positive to negative. The spatial pattern of the negative regression coefficient in the west of the GZUA showed a decrease at first but then an increasing trend.

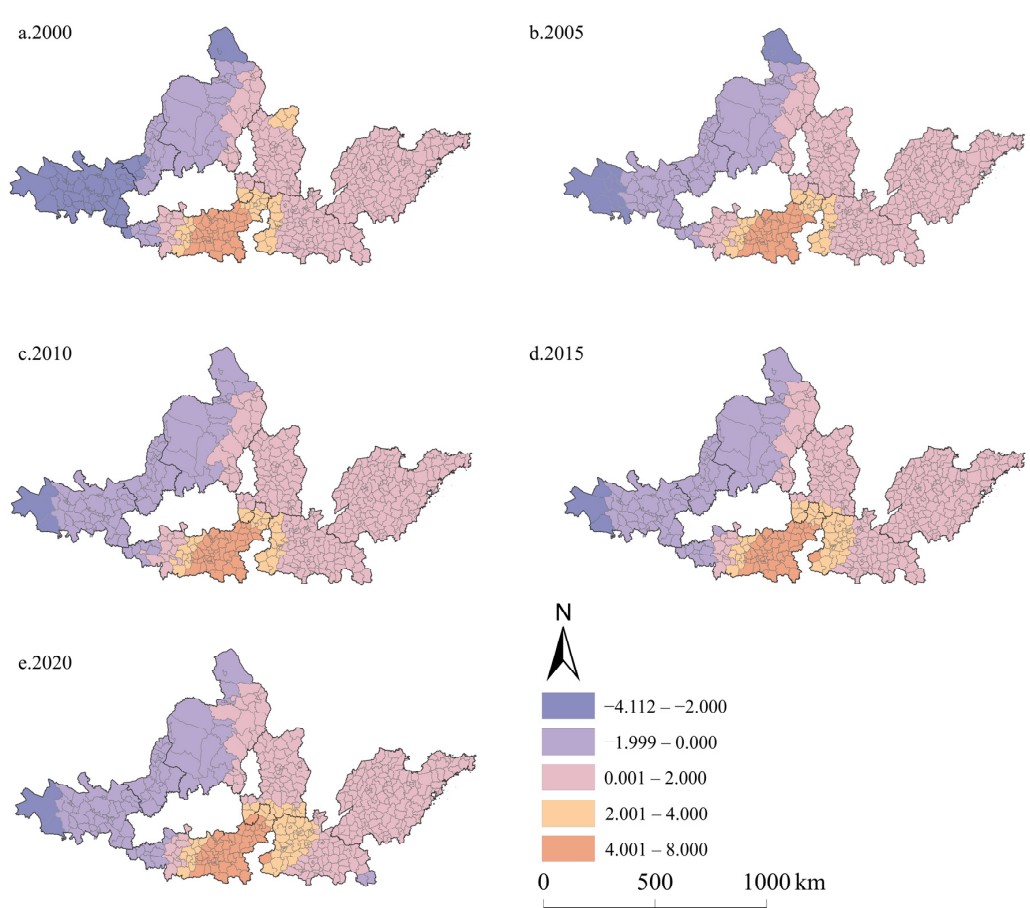

**Figure 8.** Regression coefficient of UE on CEI in the UAYRB during 2000–2020. (**a–e**) are the regression coefficient of UE on CEI in the UAYRB in 2000, 2005, 2010, 2015, and 2020, respectively.

The regression coefficients of different urban landscape metrics on CEI in the UAYRB varied greatly (Figure 9). Except for the LSI, the overall regression coefficients of the remaining three landscape metrics were positive. The regression coefficients of the AI on CEI in 2000, 2005, 2010, 2015, and 2020 were 14.500, 18.821, 22.325, 24.808, and 27.128, respectively, showing a growing trend. Spatially, similar to the distribution pattern of the regression coefficient of UE on CEI, the CPUA and SPUA concentrated the high regression coefficient. The proportion of the low regression coefficient in the west showed a tendency to shrink. The regression coefficients of the IJI on CEI in 2000, 2005, 2010, 2015, and 2020 were 0.358, 0.433, 0.475, 0.512, and 0.528, respectively, showing an increasing trend. However, the regression coefficients in some areas were negative, concentrated in the east

of the SPUA and the whole GZUA. It is noteworthy that the regression coefficients in the whole GZUA was positive in 2000. Meanwhile, the spatial distribution of negative regression coefficients in the east of the SPUA and the north of the HBOYUA narrowed over time. The regression coefficient of PD on CEI showed a steady downward trend during the study period, but remained positive. However, its spatial distribution pattern changed significantly. The regression coefficient in the west of the UAYRB was generally higher than that in the east. The south of the CPUA remained the region with a low regression coefficient while the southwest of the NXUA remained the region with a high regression coefficient. The regression coefficient of the LSI on CEI remained negative. The most significant changes happened in the GZUA, while the remaining regions almost stayed the same with limited changes. Regions with the lowest regression coefficient tended to narrow while the high regression coefficient expanded to the east of the GZUA and the southwest of the CPUA.

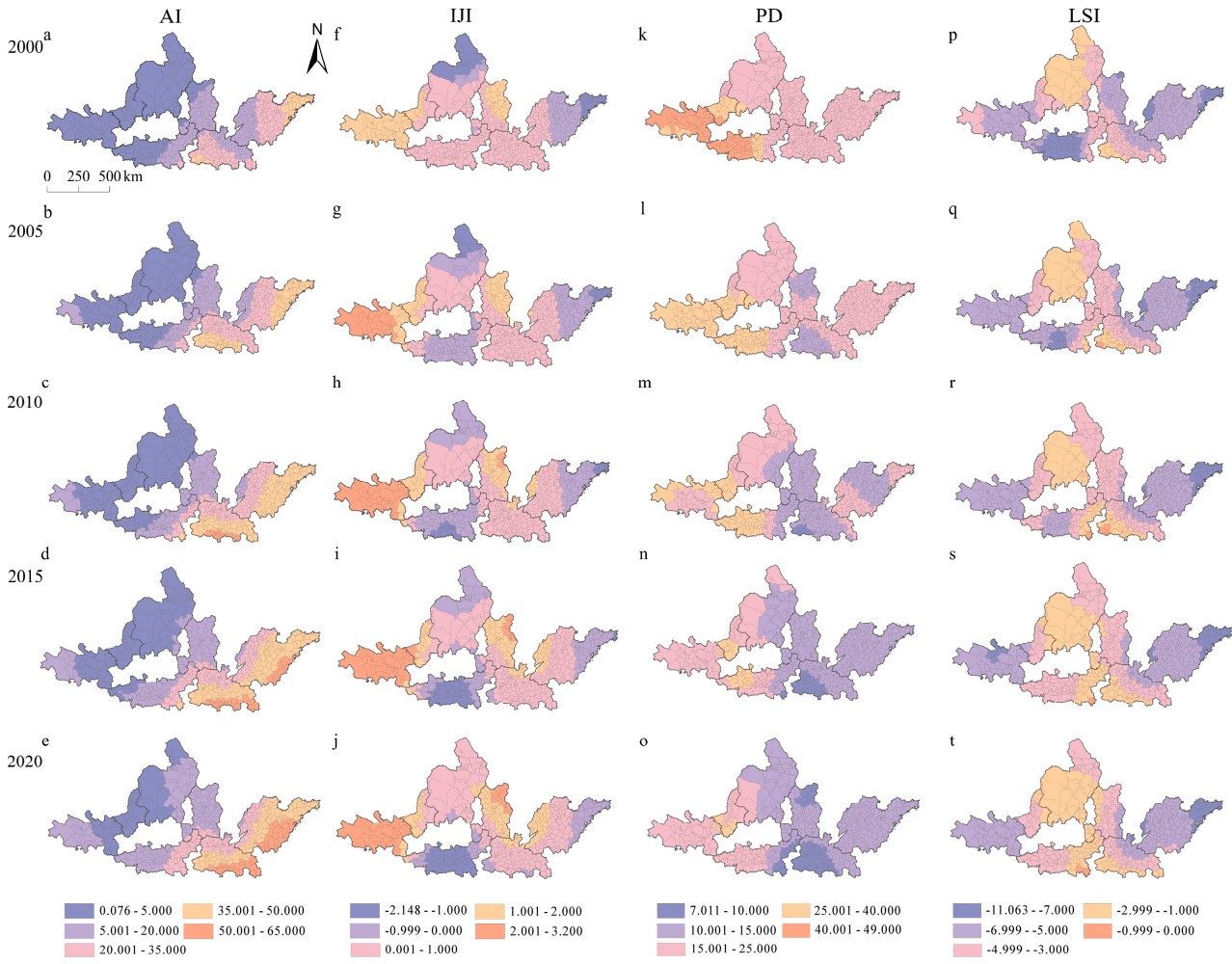

**Figure 9.** Regression coefficient of urban landscape metrics on CEI in the UAYRB during 2000–2020.

## 4. Discussion

### 4.1. Comparing with Previous Findings

Compared with previous studies, this study explored the impact of UE on CEs more comprehensively. It filled the research gap on the UAYRB, a region where current related studies have focused on specific scale units, such as global [53], national [54,55], provincial [56], and prefectural [57–60]. And, fewer scholars have studied the impact of UE on CEI using the county as a unit of study in the UAYRB. Some scholars also found that the CEs of the UAYRB increased sharply in general from 2000 to 2020 [61,62],

and showed a pattern of high in the middle-east and low in the northwest [63], and that UE exacerbated CEs [53], which was in line with the results of this study. In addition, the urban spatial pattern influences the CEs status [64], and existing studies have focused on three aspects of urban landscape patterns: urban sprawl, urban form complexity, and urban compactness. It has been recognized that urban form (the spatial pattern and structural features of urban land use) is related to urban CEs; nevertheless, there are only a few studies that have empirically assessed the direct impacts and their relationship with CEs. Fang et al. (2015) found a positive association between the growth of urban areas and $CO_2$ in 30 provincial capitals in China, that an increase in urban continuity had a dampening effect on CEs, that an increase in the complexity of urban shapes positively affected CEs, that measures to make China's existing cities more compact may actually help to reduce CEs, and that increased fragmentation or irregularity of urban form may lead to increased CEs [60]. Ou et al. (2013) quantitatively analyzed the aspects of the impact of different urban shapes on CEs and found that increased fragmentation or irregularity of urban shapes may also lead to more CEs and that a compact development pattern of urban land use helps to reduce CEs [59]. The regression results of this study were the similar as those of the above studies. It was found that the regression coefficient between the LSI and CEI was negative, indicating that there was a negative association between the LSI and CEI, and the larger the value of the LSI, the more complex the landscape shape and the fewer CEs. This suggests that increased continuity in the urban landscape has a dampening effect on CEs. The positive regression coefficient between the AI and CEI indicates that CEs increase as urban form complexity increases. High LSI values may indicate a higher complexity of landscape patches, representing a richer ecosystem or green space environment within the city, helping to absorb and fix a large amount of CEs, thereby reducing CEs [65]. High AI values may indicate that urban interior landscape patches are more closely connected, reflecting the increasing degree of urbanization, resulting in more intensive and convenient transportation, concentration of industrial activities and intensification of land use, which may lead to more transportation emissions and energy consumption, as well as more industrial and commercial consumer electronics, thereby increasing CEs [66]. In addition, the positive regression coefficients between PD and the IJI and CEI also indicate that the increase in these urban landscape metrics may lead to increased CEs.

*4.2. Spatiotemporal Differences in the Impact of UE on CEI*

UE is a prevailing phenomenon in most parts of the world [67]. It is projected that urban areas are expected to triple between 2000 and 2030 [3]. Corresponding to UE, CEs in the past century have also witnessed a rocketing rise [68], leading to an increase in global average surface temperature [69]. It is estimated that such a global warming phenomenon will continue in the next century [70]. Urban areas account for 71% of global CEs, while that will increase to 76% by 2030 [71]. However, the situation in China is even more severe. It is reported that 84% of China's commercial energy usage comes from urban areas [72], while every 1% increase in urbanization in China is associated with a 1% increase in CEs [73,74]. Although research found that some counties in China have achieved the decoupling relationship of urbanization and CEs, such a status is not stable [75]. Much has been carried out to reveal the eco-environmental effects of UE. However, much less is known about the spatial heterogeneity of the impact of UE on CEs.

This study analyzed the impact of UE on CEs and their spatial heterogeneity in the UAYRB, China. The results show that both the CEI and UE of the UAYRB showed an obvious increasing trend during the study period. The spatial distribution of them also showed similar distribution patterns, both concentrated in the SPUA and CPUA. These two urban agglomerations have developed relatively rapidly, and their favorable location and vast plain area enable them to take the lead in the entire Yellow River Basin. Moreover, the kernel density contours of the normalized CEI and UE revealed the weakening of CEI

and the potential for large-scale UE in some counties in the UAYRB. Xing et al. (2022) found the same evolution trend of CEI and a similar kind of UE in the urban agglomeration in south-central Liaoning province [76]. They found that, from 2010 to 2016, the average annual expansion rate of the urban agglomerations in south-central Liaoning province increased from 3.93% to 5.8%, with the expansion intensity increased from 0.211 to 0.525 [76]. However, the UE rate in the UAYRB region continuously slowed down. That is also the case in the Changsha–Zhuzhou–Xiangtan Urban Agglomerations [77–79]. Tian and Zhao (2024) found that UE in the Changsha–Zhuzhou–Xiangtan Urban Agglomerations decreased in 2015–2020 compared to the previous period [79]. Meanwhile, the finding in the three coastal agglomerations in China by Wen et al. (2019) also confirmed that UE is a major trend in the development of urban agglomerations [80]. Generally, affected by differences in urban resource endowment, development mode, and development direction, the UE and CEs of different cities also differ in the development process, showing spatial imbalance among regions. With the advance of time, the degree of CEs will change and the evolution trajectory of different cities will also change with different development processes. Meanwhile, in this study, the Gini coefficients of both CEI and UE were at high levels, indicating an obvious spatial imbalance of CEI and UE in the UAYRB region. It is noteworthy that UE has gradually taken on a regionally balanced pattern in the UAYRB during the study period, which can be partly attributed to the implementation of some development strategies in China. Such a phenomenon can also be found in the Changsha–Zhuzhou–Xiangtan Urban Agglomerations [79]. After the proposal of a "two-type society" in 2007, the CEs reduction of urban land in urban agglomerations has gradually achieved remarkable results [81]. Meanwhile, since the 18th National People's Congress, "ecological civilization construction" has been carried out, carbon-saving and emission reduction work have been continuously promoted, and the CEs of all districts and counties have been effectively controlled.

The synergistic relationship between UE and CEs has been confirmed in many studies. Tian and Zhao (2024) adopted Pearson correlation analysis and found that there was a high positive correlation between urban land area and CEs; that is, an increase in urban land area would lead to a simultaneous increase in CEs [79]. Xing et al. (2022) also confirmed the same synergistic expansion state of UE and CEs [80]. However, such synergies will change over time. As shown in this study, the Markov chain of CEI and UE revealed the slowing growth rate of CEI, while UE remained a continuing increasing process. This also proves to some extent that the effect of emission reduction policies implementation may be beginning to show. However, the spatial distribution of the impacts of UE on CEI may reveal that such policy effects may vary greatly among regions. In areas where UE and CEI values are high, the impacts of UE on CEI remain mostly positive with a lower regression coefficient, while, in places where UE and CEI values are low, the impacts of UE on CEI remain mostly negative with a higher regression coefficient. The difference in the response degree to the policy, the intensity of implementation, and the level of technology in different regions are the main reasons for the difference in the effectiveness of policy implementation. Areas with a relatively backward economic level, such as the LXUA and NXUA, have policies that restrict economic development but favor environmental protection that are often not well implemented. Meanwhile, policies are often ineffective when implemented due to the insufficient level of technology to support the low-carbon development of energy-consuming industries in the urbanization process in these regions. Therefore, the spatiotemporal differences in the impact of UE on CEs can be quite different, leading to significant spatial differences in the high-quality development level of the UAYRB, with the high-quality development index in the middle and lower reaches higher than that in the middle and upper reaches [82]. This conclusion is unanimously supported by Fang (2020) and Sun et al. (2022). Fang (2020) found that the higher upstream the Yellow River Basin, the lower the development of urban agglomerations [40], while Sun et al. (2022) revealed that the coordination degree of the upstream urban agglomerations is lower than that in the middle and lower reaches [39].

### 4.3. Policy Implications

Within the context of dual-carbon goals and new urbanization, ensuring stable economic growth while reducing CEs is one of the most important tasks for achieving high-quality urban development. UE is one of the most important manifestations of urban agglomerations development, and a reduction in the level of CEs in response to UE is conducive to the low-carbon development of urban agglomerations [79]. This study analyzed the impact of UE on CEs in the UAYRB region and proposed policy recommendations for low-carbon development in the following areas.

Although the UE rate in the UAYRB has slowed down, it is still in the expanding progress and some counties have the potential for large-scale UE. Once the process is out of control, it will inevitably lead to an unnecessary waste of resources and inefficient use of land. The expansion of urban land may also bring the possibility of occupying cultivated land, forest land, and other ecological land, which further aggravates the ecological and environmental problems of the city [83]. Therefore, it is highly recommended to formulate relevant legal policies to strictly control the urban sprawl. Meanwhile, strong penalties for illegal sprawl should be applied to enforce the policy to the end. In addition, urban planning is an important means of limiting urban sprawl [84], and construction land development needs to be carried out in strict accordance with the requirements of urban planning to ensure the sustainable development of cities. Local governments should abandon the idea of land finance and prioritize the healthy governance of cities and the realization of green and sustainable development to meet the needs of the people's happy lives [85].

Due to the needs of economic development, some regions are still at a stage of high CEs. In addition, although relevant energy-saving and emission reduction measures have been taken, they have not achieved the desired effect of emission reduction. For these areas, land resources should be used economically and intensively to avoid the disorderly expansion of urban land in the study area and realize the compact development of urban space. The proportion of tertiary industry should be increased, some high-polluting industries should be eliminated, advanced technologies should be introduced to improve CEs efficiency, and research and development of key technologies should be accelerated to achieve CEs reduction. At the same time, green areas should be increased to achieve carbon sink increase so as to achieve a strong decoupling as soon as possible and to enter into the stage of a low-carbon economy.

According to the spatial pattern of urban land use in the urban agglomerations of the UAYRB from 2000 to 2020 in the results of the study, including the AI, IJI, PD, and LSI, first of all, the overall decreasing trend of the AI indicates that the urban landscape of the UAYRB as a whole shows a certain degree of a discrete trend during the study period, which may be related to factors such as UE and land use changes. Second, the spatial pattern of the IJI shows a tendency to expand to the east. PD and the LSI continued to grow during the study period. According to the regression coefficients of urban landscape indicators on CEI in the research results, we can focus on areas with high regression coefficients, prioritize the formulation of targeted environmental protection policies and measures to cope with the trend of their growth in urban landscape indices, and design targeted environmental protection programs according to the distribution of the regression coefficients in different areas. For example, ecological protection and greening can be strengthened in CPUA and SPUA areas. In the process of policy formulation, it is necessary to consider the trends and spatial distribution patterns of the above indicators, strengthen the ecological protection and restoration of urban fringe areas, optimize the urban layout, and rationally guide urban development and low-carbon transformation.

### 4.4. Limitation and Future Directions

Although we tried to reveal the impact of UE on CEs in UAYRB with a series of analysis models, certain limitations still exist in this research. First of all, this study only analyzed the impact of UE on CEI, while the impacts of UE on CEs per GDP, CEs per person, or total CEs were still not clear. Different ways of quantifying CEs may produce more

pronounced differences in results. In addition, this study explored the impact of the urban landscape pattern index on CEs with the GTWR model, while the internal mechanism of the impact of UE on CEs has not yet been clarified. Meanwhile, in this study, the impact of UE on CEs was analyzed only at the county scale, while a smaller scale can capture more regional differences. Therefore, the next step can be to conduct a smaller-scale analysis based on improved data accuracy to capture more features of regional differences. These limitations point to future research directions.

## 5. Conclusions

In this study, we analyzed the UE patterns and CEI in the UAYRB at the county level with kernel density, the Gini coefficient, and Markov chains, and revealed the spatial heterogeneity of the impact of UE on CEI with the GTWR model. The results show that CEI and UE in the UAYRB both showed a steady growing trend during the study period. The kernel density contours of the normalized CEI and UE revealed that the region with high values of CEI was continuously shifting to lower values, while the UE rate continuously slowed down in the UAYRB region. The Gini coefficients of both CEI and UE were at high levels, indicating obvious spatial imbalance in the UAYRB region. The Markov transfer probability matrix for CEI with a time span of five years showed that CEI growth will still occur over the next five years, while that of UE was more obvious, indicating the existence of the probability that larger-scale UE will still occur in the next five years. Counties with a regression coefficient of UE on CEI higher than 0 covered the majority, remaining as having a relatively stable distribution pattern. The regression coefficients of different urban landscape metrics on CEI in the UAYRB varied greatly; except for the LSI, the regression coefficients of the AI, IJI, and LSI remained positive. The study findings can enlighten policy implications for the high-quality development of the Yellow River Basin. The above analysis showed that, during the study period, CEs and UE in the UAYRB have continued to increase, showing obvious uneven spatial distribution. Looking forward to the next five years, the trend of CEs and UE will continue to grow, while the growth rate of CEs will gradually slow down and UE may be further accelerated in the following decade. The urban landscape showed a clear trend of discretization. Based on the urban landscape index and regression results, the characteristics of each region of the UAYRB must be fully considered in order to rationally guide urban development paths and realize low-carbon transformation.

**Author Contributions:** Conceptualization, Z.W. and W.C.; methodology, Z.W., Y.Z., T.G., X.W. and W.C.; software, Y.Z., T.G. and X.W.; validation, Z.W. and W.C.; formal analysis, Z.W. and W.C.; investigation, Z.W. and W.C.; resources, Z.W. and W.C.; data curation, Y.Z., T.G. and X.W.; writing—original draft preparation, Y.Z., T.G. and X.W.; writing—review and editing, Y.Z., T.G. and X.W.; visualization, Y.Z., T.G. and X.W.; supervision, Z.W. and W.C.; project administration, Z.W. and W.C.; funding acquisition, Z.W. and W.C. All authors have read and agreed to the published version of the manuscript.

**Funding:** This research was supported by the Ministry of education of Humanities and Social Science project (Granted No. 19YJC630179).

**Data Availability Statement:** The data that support the findings of this study are available from the corresponding author upon reasonable request. The data are not publicly available due to privacy restrictions.

**Acknowledgments:** The authors would like to thank the anonymous reviewers for their constructive comments on improving this paper.

**Conflicts of Interest:** The authors declare no conflicts of interest.

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
