# Peer review of "Impact of Urban Expansion on Carbon Emissions in the Urban Agglomerations of Yellow River Basin, China"

_land, doi:10.3390/land13050651_

Round 1

Reviewer 1 Report

Comments and Suggestions for Authors

Dear Authors, 

Thanks a lot for your well-prepared manuscript. However, I have a minor concern regarding lacking a sample of introduced data that is used to come out with your analysis and results. It is highly recommended to show some of your data in a table in the methodology section to get the attention of the readers.

My best regards 

Author Response

Thanks a lot for your well-prepared manuscript. However, I have a minor concern regarding lacking a sample of introduced data that is used to come out with your analysis and results.

Comment 1: It is highly recommended to show some of your data in a table in the methodology section to get the attention of the readers.

Reply: Thank you very much for your suggestion. Based on your suggestion, we have explained our data in more detail in the data source section. (Page 4).

2.2. Data sources

The data in this study involved CEs and land use data. Source of CEs data from the open-source data inventory (https://db.cger.nies.go.jp/dataset/ODIAC/) [41], this CEs da-ta successfully estimates the spatial distribution of fossil fuel CEs on a global scale by combining nighttime lighting data and emission location profiles of individual power plants using an innovative emission modeling approach. The spatial resolution is 1000 m. The land use change data was downloaded from the Data Center for Resources and Environmental Sciences and the Chinese Academy of Sciences (https://www.resdc.cn/). This study used the proportion of urban land and the landscape pattern index to characterize UE [42]. The proportion of urban land is the ratio of urban land area to the area of the study unit, and the landscape pattern indexes were calculated in Fragstats v4.2.1.

Reviewer 2 Report

Comments and Suggestions for Authors

The manuscript entitled "Impact of urban expansion on carbon emissions in the urban agglomerations of Yellow River Basin, China" presents a good and interesting work. In general, the manuscript should be acceptable for publication but some problems could be repaired prior to publication. Some suggestions are as follows:

1. The text of the "introduction" does not sufficiently develop the function of this type of research for this field of knowledge. The systematization of metological procedures should be described in more detail.  Is this proposal inedited? If so, emphasize the unpublished. If not, cite previous work discussing this method integration; 

2. In Introduction, the novelty, economic impact and practical applicability of this study should be highlighted more; 

3. Correct references in the text and the reference list according to the journal's format;

4. A proper presentation of the study area is not performed. Even the location map does not present, for example, in which region of Asia the study area is located;

5. You could enrich the scientific literature. Should be included some bibliographic references and and a deeper discussion;

6. A basic flowchart of the suggested methodology should be presented in the paper. Thus, the readers can easily follow the application procedures;

7. All maps have low resolution. It is not possible to properly identify any of the subtitles;

8. Conclusions has only the main results in a summary. But what were the main findings and advances of this research clearly?  

Reviewer 3 Report

Comments and Suggestions for Authors

The paper deals with an interesting and urgent issue related to the carbon emission intensity distribution with a particular focus on the yellow river basin in China. Considering the increasing trend in land use in the Chinese context, the study can contribute to understanding which measures can be taken to reduce impact in a fragile environment.

The introductory section provides a clear a referenced framework picture both describing the specific regional conditions and the previous studies on the topic pointing out both the consolidated approaches and the remaining gaps.

Section 2 material and methods need to be strengthened clearly stating which is the scope of the study and why a certain methodological approach has been developed linking the performed tasks to the specific expected outcomes. At the moment, this section simply escribes the subject and the variables without explaining the decisional process and the why certain tools/methods are adopted with relation to the achievement of the final objective.  A workflow diagram would certainly help the visualization and the understanding of the process overall. Additionally, the description of the study area (2.1) must follow the general description of the method which is supposed to be applicable in any potential case study area. It would also be useful to explain why Kernel density, Gini coefficient, Markov chain were adopted and which the expected contribution to the overall analysis.

In the result section, the role of the landscape metrics of aggregation index (AI), interspersion and juxtaposition index (IJI), patch density (PD), and landscape shape index (LSI) should be introduced with relation to the expected results and the methodological structure.

The results are critically discussed however more punctual observations related to the outcomes of the diagrams would help in clarifying the value of the performed analysis. The study provides a reflection on the policy implications however it misses to provide concrete indications or strict parameters to address future growth leaving the discussion mostly on the theoretical level while the current trends call for urgent actions to more environmental-friendly solutions. This is a shortcoming. The limitations of the conducted study are fairly mentioned within the discussion and the conclusions.

Comments on the Quality of English Language

English language is generally satisfactory, a final reading to solve typos is suggested.

Round 2

Reviewer 2 Report

Comments and Suggestions for Authors

Dear Authors,

You made all the required revisions and qualifically justified my questions. Therefore, my decision is to "accept".

I am available for any clarification.

Kind regards.

Reviewer 3 Report

Comments and Suggestions for Authors

The authors put a certain effort to address the reviewer’s remarks and suggestions, improving the quality of the paper overall. The method section has been clarified and supported by a workflow diagram which makes the process more understandable both in its scope and expected outcomes.

Additional comments and explanations have been integrated in the result and discussion sessions.

Comments on the Quality of English Language

English language is generally satisfactory, a final reading to solve typos is suggested.